# Friction modulation in limbless, three-dimensional gaits and heterogeneous terrains

Xiaotian Zhang[1], Noel Naughton [1,2], Tejaswin Parthasarathy[1] & Mattia Gazzola [1,3,4,5]✉

Motivated by a possible convergence of terrestrial limbless locomotion strategies ultimately determined by interfacial effects, we show how both 3D gait alterations and locomotory adaptations to heterogeneous terrains can be understood through the lens of local friction modulation. Via an effective-friction modeling approach, compounded by 3D simulations, the emergence and disappearance of a range of locomotory behaviors observed in nature is systematically explained in relation to inhabited environments. Our approach also simplifies the treatment of terrain heterogeneity, whereby even solid obstacles may be seen as high friction regions, which we confirm against experiments of snakes 'diffracting' while traversing rows of posts, similar to optical waves. We further this optic analogy by illustrating snake refraction, reflection and lens focusing. We use these insights to engineer surface friction patterns and demonstrate passive snake navigation in complex topographies. Overall, our study outlines a unified view that connects active and passive 3D mechanics with heterogeneous interfacial effects to explain a broad set of biological observations, and potentially inspire engineering design.

[1] Department of Mechanical Science and Engineering, University of Illinois at Urbana-Chmpaign, Urbana, IL 61801, USA. [2] Beckman Institute for Advanced Science and Technology, University of Illinois at Urbana-Champaign, Urbana, IL 61801, USA. [3] National Center for Supercomputing Applications, University of Illinois at Urbana-Champaign, Urbana, IL 61801, USA. [4] Carl R. Woese Institute for Genomic Biology, University of Illinois at Urbana-Champaign, Urbana, IL 61801, USA. [5] Center for Artificial Intelligence Innovation, University of Illinois at Urbana-Champaign, Urbana, IL 61801, USA. ✉email: mgazzola@illinois.edu

Limbless locomotion is exhibited by a wide taxonomic range of slender creatures and has been observed in water[1], land[2–5], and even air[6]. While broad principles of aquatic limbless locomotion have been unveiled[1,7–11], the terrestrial variety remains largely elusive. In snakes, locomotion has been classically modeled via planar gaits on uniform substrates, with lateral body undulations rectified into forward motion via anisotropic friction[12–20]. However, terrestrial creatures (unlike aquatic ones) can actively negotiate the extent of contact with the environment, by lifting selected body regions. This manifests in a variety of nonplanar, transient, and spatially inhomogeneous gaits whose locomotory outputs, in turn, emerge from the interplay with dirt, sand, mud, rocks, or leaves[2,21–25], typical of environments that are nonuniform and themselves poorly physically understood.

Despite these challenges, a recent convergence between zoologists, physicists, mathematicians, and roboticists has provided new impetus toward understanding the organization of out-of-plane behaviors[13,21,26–29]. Among these, particular attention has been devoted to sidewinding (Fig. 1a), whereby snakes can travel at an angle to overall body pose and reorient with neither loss of performance nor kinematic precursors—features that render sidewinders economical, elusive, and versatile dwellers[30]. Long puzzling scientists, sidewinding has been recently recapitulated in robot replicas by means of simple actuation templates made of two orthogonal body waves[26], demonstrating steering abilities and ascending of sandy slopes[27].

Although insightful, experimental approaches are specialized to given animal/robotic models and there is still a noticeable lack of a broader theoretical perspective able to relate the interplay between gait (body deformation) and frictional environment to locomotory output emergence. Here, motivated by a possible evolutionary convergence of limbless movements ultimately determined by interfacial effects, the roles of both 3D body deformations and environmental heterogeneities are connected through, and modeled as, planar friction modulations. Thus, by homogenizing the complex interaction between limbless creatures and substrate features into a spatially and temporally varying 2D frictional field, we combine theory and simulations to establish an effective-friction perspective that coherently explains a broad set of observations.

## Results

**Theoretical modeling approach.** To gain mechanistic insight, we generalize a model of forward slithering, first proposed by Hu and Shelley[13,14], to encompass a richer variety of behaviors. The model instantiates a snake as a 2D planar curve of lateral curvature $\kappa(s,t) = \epsilon \cos(2\pi k(s+t))$ with arc-length $s \in [0,1]$, time $t$, amplitude $\epsilon$, and wavenumber $k$, from which midline positions $\mathbf{x}(s,t)$ and orientations $\alpha(s,t)$ follow (Fig. 1c, Methods). For all quantities, space is scaled on snake's length $L$ and time on wave propagation period $\tau$. Net propulsion forces $\mathbf{F}_{\text{net}}$ and torques $\mathbf{T}_{\text{net}}$, obtained by integrating friction forces over body length and period, propel the snake. Anisotropic friction forces are described via the Coulomb model $\mathbf{F}(s,t) = -N(s,t)\boldsymbol{\mu}$, where $\boldsymbol{\mu}$ is a function of forward $\mu_f$, transverse $\mu_t$, and backward $\mu_b$ friction coefficients (Fig. 1c) capturing the interaction between skin texture and substrate[13]. Of these, $\mu_b$ has little effect[12,14], leaving $\mu_t/\mu_f$ as the key characteristic parameter. Thus, system dynamics are governed by the ratio of inertia to friction forces, via the Froude number $\text{Fr} = (L/\tau^2)/(g\mu_f)$, with $g$ being gravitational acceleration. In biological and robotic snakes, friction typically dominates with $\text{Fr} \leq 1$, and we set $\text{Fr} = 0.1$ throughout, without lack of generality (SI). Finally, the function $N(s,t) = \eta \hat{N}(s,t)$, with $\eta$ being a normalization factor, models body lift as local weight redistribution, leading to effective-friction modulation along the snake. This modulation implicitly gives rise to a temporally and spatially varying field of frictional force magnitudes, actively controlled by the snake.

The term $\hat{N}(s,t)$ is critical, and much of the model's explanatory power depends on it. Hu et al.[13,14] set $\hat{N} \sim e^{-\kappa}$ to

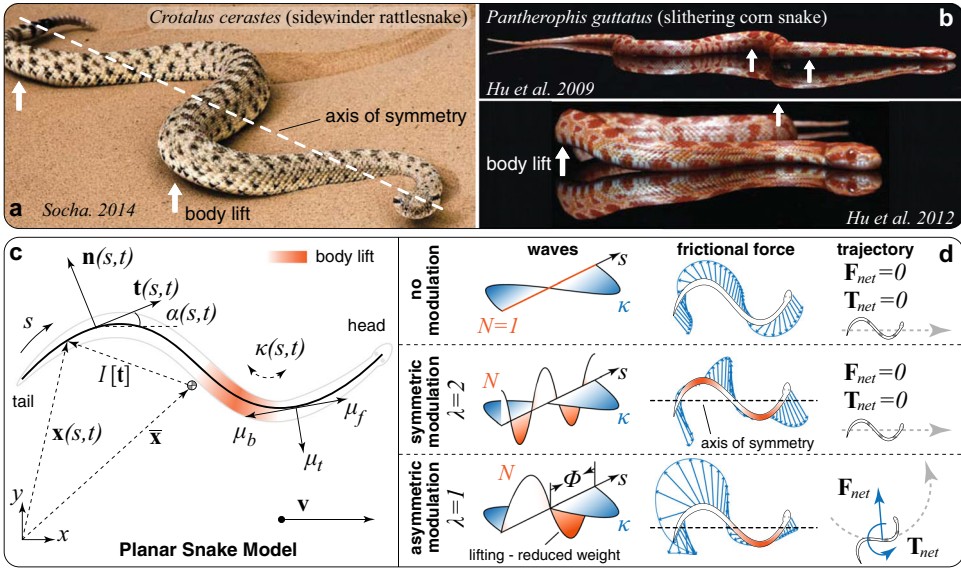

**Fig. 1 Examples of biological snakes employing a lifting body wave in addition to lateral undulation. a** A sidewinding rattlesnake (*Crotalus cerastes*) asymmetrically lifts up only one side of its body[52]. **b** A corn snake (*Pantherophis guttatus*) slithering on a flat surface and symmetrically lifting regions of high lateral curvature on both sides of its body[13, 14]. **c** Schematic of the planar snake model. Note that the arc-length *s* goes from tail to head to retain consistency with[13, 14]. The local position **x** is related to the center of mass **x̄** through zero-mean integration function *I*[**t**] (Methods). **d** Three different stereotypes of body lifting. Top: Zero body lifting leads to classical undulatory planar gaits. Middle: Symmetric body lifting, the snake symmetrically lifts both sides of its body[13, 21]. Bottom: Asymmetric body lifting, the snake lifts one side of its body off the ground and maintains the other in contact with the ground. Asymmetric lifting has been well-documented in sidewinding snakes[2, 21, 22, 26, 27]. Net forces and torques acting on the snake over one undulation period are computed via $\mathbf{F}_{\text{net}} = \int_0^1 \int_0^1 \mathbf{F}(s,t)\,ds\,dt$ and $\mathbf{T}_{\text{net}} = \int_0^1 \int_0^1 (\mathbf{x} - \bar{\mathbf{x}}) \times \mathbf{F}(s,t)\,ds\,dt$, respectively.

capture lifting effects at regions of high lateral body curvature in forward slithering snakes (Fig. 1b), demonstrating drag reduction and speed increase. Nonetheless, this choice does not capitalize on the opportunity of temporally decoupling lateral and lifting activations, to break symmetry and allow the investigation of locomotory outputs other than forward slithering. While a variety of functions $\hat{N}$ can achieve this (SI), a phase shift in a cosine form consistent with lateral curvature is perhaps the simplest and most natural option. Thus, here we set $\hat{N}(s,t) = \max\{0, A\cos(2\pi k_l(s + t + \Phi)) + 1\}$ where $A$ is lifting amplitude, $\Phi$ is phase offset with lateral wave $\kappa$, and $\lambda = k_l/k$ is the ratio of lateral to lifting wave numbers. The max function avoids artificial negative weight redistributions.

This parameterization allows us to model and compare stereotypical lifting patterns encountered in nature (Fig. 1d). For $A = 0$, classic planar undulatory gaits are recovered[13,23]. For $|A| > 0$ and $\lambda = 2$, the snake symmetrically lifts both sides of its body, as in[13,21]. In both cases, $\mathbf{F}_{\text{net}} = \mathbf{T}_{\text{net}} = 0$ due to symmetry and the snake can only move forward. If instead $\lambda = 1$, the snake lifts only on one side, as seen in sidewinders[23,27]. This breaks friction forces symmetry, allowing maneuvering ($\mathbf{F}_{\text{net}}$, $\mathbf{T}_{\text{net}} \neq 0$) without changes in the lateral gait $\kappa$.

**Emergence of locomotory behaviors in context with the environment.** To investigate the potential of $\lambda = 1$ lifting waves for locomotion, we identify the behaviors available to a snake in relation to its frictional environment. We consider first the ratio $\mu_t/\mu_f = 2$, which captures the frictional interaction between anisotropic scales and firm uniform substrates, determined for anesthetized snakes[13]. Since snakes actively control their scales for grip[15], $\mu_t/\mu_f = 2$ may be considered a lower bound estimate.

We numerically span the $A$–$\Phi$ plane and characterize locomotory outputs by steering rate $\dot{\theta}$ and body pose $\gamma$ (Fig. 2a, b), based on experimentally observed behaviors[2,21,22,26,27]. Key organizing separatrices emerge (Fig. 2c). Along $A = 0$ or $\Phi \sim 1/4$ and $3/4$, the snake can only travel in rectilinear trajectories ($\dot{\theta} = 0$), whether it is slithering or sidewinding. At the same time, along $A = 0$ or $\Phi \sim 0$ and $1/2$, the snake is always tangent to its trajectory ($\gamma = 0$), whether traveling rectilinearly or turning. Around this underlying structure, locomotion behaviors naturally organize as phases (Fig. 2d). Straight slithering is encountered throughout $\Phi$ for small $A$, with limited turning abilities observed in small regions at $\Phi \sim 0$ and $1/2$. Sidewinding clusters around $\Phi \sim 1/4$ and $3/4$ for larger lifting. This explains observations of $\Phi \sim 1/4$ in biological sidewinders[22,23] and empirical robotic demonstrations[26,27]; indeed only in the neighborhood of this particular offset (or equivalently $\Phi \sim 3/4$) can both linear trajectories and large pose angles co-exist. Finally, spinning in-place[26] fills gaps at high liftings.

To further contextualize these findings, it is useful to investigate how changes in skin texture–environment interaction, captured by $\mu_t/\mu_f$, affect phase space organization. As we vary $0.5 < \mu_t/\mu_f < 10$ in Fig. 3a, separatrices are approximately retained, while behavioral outputs drastically remodel, appear, and disappear. For example, for $\mu_t/\mu_f < 1$ (a condition not commonly encountered in nature, included here for completeness) slithering is replaced by a new, backward counterpart wherein snakes completely reverse their travel direction.

For isotropic friction $\mu_t/\mu_f = 1$, planar ($A = 0$), and asymmetric lifting ($\lambda = 1$) slithering are no longer available, and snakes must instead either sidewind or switch to symmetric lifting ($\lambda = 2$) for locomotion. However, sidewinding is found to be significantly faster (Fig. 3b), thus proving advantageous in environments, such as sandy deserts or mudflats, characterized by low effective-

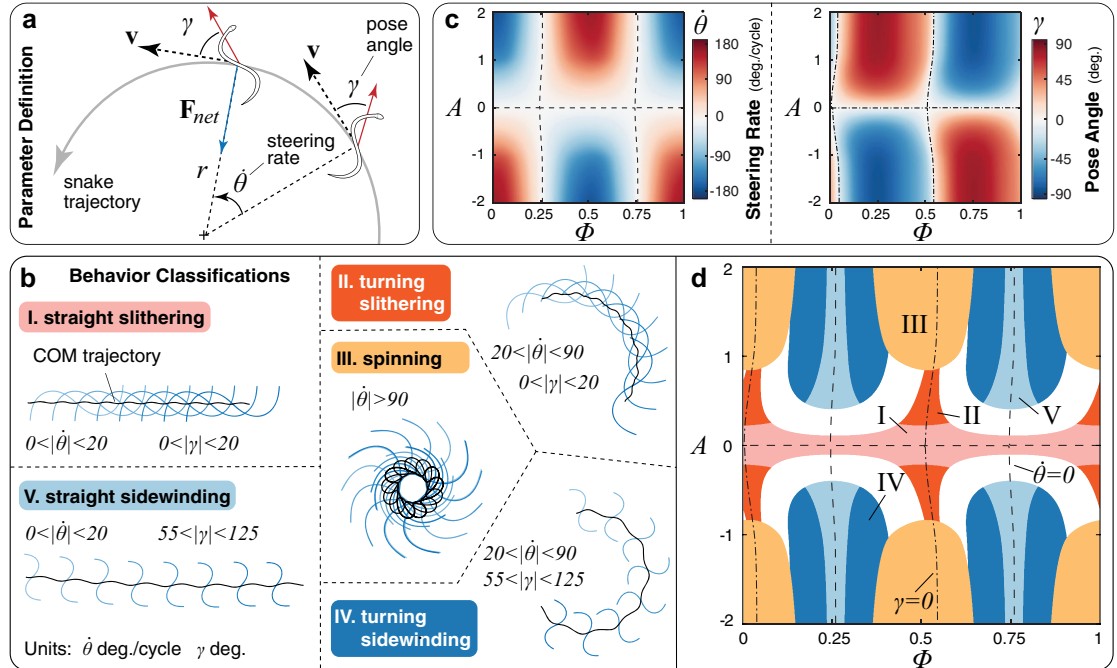

**Fig. 2 Locomotory behaviors available to a snake as function of body lifting $A$ and phase offset $\Phi$. a** Quantities used to analyze locomotion behavior. Steering rate $\dot{\theta}$ is the time-averaged angular velocity of the snake's center of mass. Pose angle $\gamma$ is the angle between the snake's orientation and velocity direction (Methods). **b** Classifications of qualitatively different locomotion behaviors given $\dot{\theta}$ and $\gamma$ (Supplementary Movie 1), based on experimentally observed pose angles of sidewinding snakes[33]. Black lines are the snake's center of mass (COM) trajectories. **c** Field map of steering rate $\dot{\theta}$ and pose angle $\gamma$ for varying lifting wave amplitude $A$ and phase offset $\Phi$ (for $\mu_t/\mu_f = 2$). **d** Phase space of locomotion behaviors available to a snake for the friction ratio $\mu_t/\mu_f = 2$. White spaces are transition regimes between different behaviors. Separatrices are zero contours for steering rate (dash line) and pose angle (dash-dot line). All simulations employ $\epsilon = 7$ and $k = 1$, as in[13].

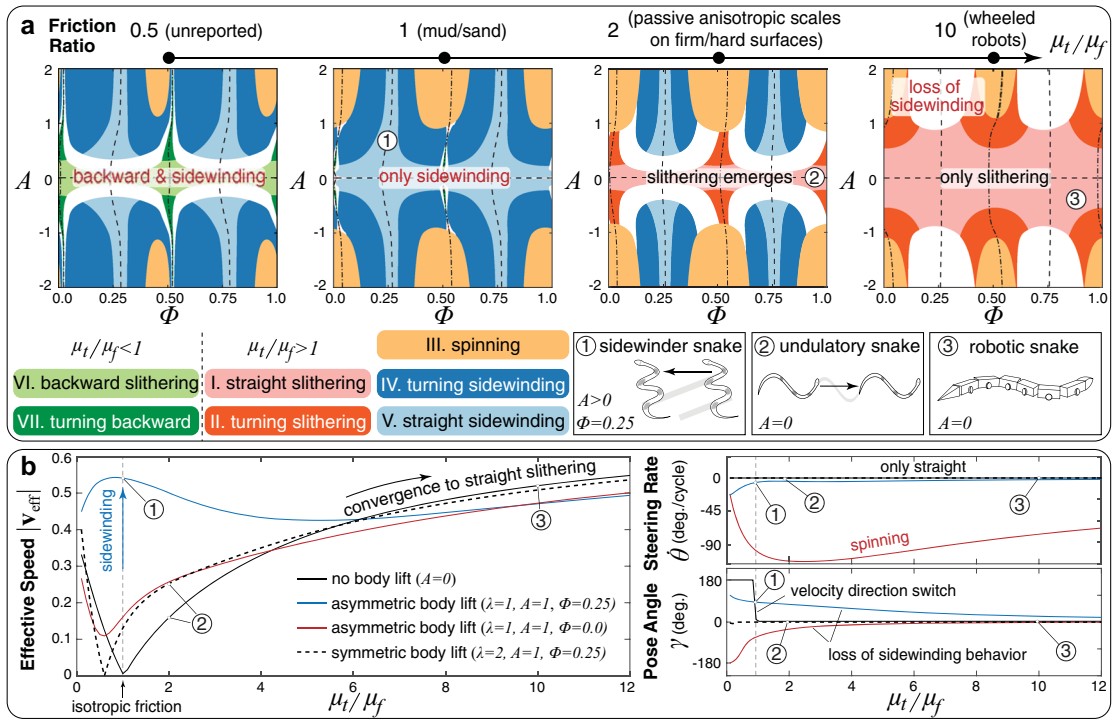

**Fig. 3 Emergence of locomotory behaviors in context with the environment – influence of friction ratio. a** Phase space maps for different friction ratios $\mu_t/\mu_f$ (full exploration in SI). Numbered labels indicate the location in terms of friction ratio, lifting amplitude, and phase offset of typical locomotion behaviors: (1) a snake sidewinding in a sandy desert or a tidal mudflat[23], (2) an undulating snake with no body lift based on measurements from[13], and (3) a wheeled robotic snake where the wheels can be viewed as inducing strong friction anisotropy. Note that for $\mu_t/\mu_f = 1$, straight sidewinding does not occur for $A = 0$ as the snake is unable to produce directional motion when there is no body lift. **b** Effective speed, steering rate, and pose angle of four different lifting strategies over a range of friction ratios. Different body lifting strategies lead to large differences in all three quantities at low friction anisotropy ratios, while there is a general convergence of behaviors to traveling forward in a straight trajectory as friction anisotropy increases. For isotropic friction, the no lifting case of $A = 0$ corresponding to planar slithering has $|\mathbf{v}_{\text{eff}}| = 0$, while asymmetric lifting with $A = 1$ and $\Phi = 0.25$, corresponding to sidewinding[26, 27], exhibits high $|\mathbf{v}_{\text{eff}}|$.

friction ratios on account of their propensity to yield under stress[31,32]. This is consistent with sidewinders inhabiting such terrains[31], while, conversely, slithering snakes are observed to adopt sidewinding when encountering sand and mud[23,25,33]. Further supporting predictions, the application on slithering snakes of cloth 'jackets' that eliminate anisotropy has been found to severely impair their locomotory performance[34].

As friction ratios increase ($\mu_t/\mu_f > 2$), we observe a progressive loss of sidewinding behavior (Fig. 3a) and a convergence toward slithering (Fig. 3b), which becomes increasingly fast and eventually, for sufficiently large values ($\mu_t/\mu_f > 10$, e.g., wheeled robots), the only option. This is consistent with the fact that sidewinders and slitherers are comparably fast in their respective habitats[14]. It is also consistent with observations that sidewinding rarely occurs outside of sandy and muddy terrains[33], although some desert sidewinding-specialists, most notably *Crotalus cerastes*, do sidewind on substrates other than isotropic sand[33]. This is not in contrast with our model, which, in fact, allows for sidewinding at moderate friction ratios, albeit at the cost of increased body lift (Fig. 3a).

Thus, by reducing out-of-plane deformations to waves of active friction modulation, our simple model coherently captures a broad set of experimental observations, providing a mapping between the gait, frictional environment, and locomotory output. In particular, it corroborates the hypothesis, never mechanically rationalized, of sidewinding being an adaptation to sandy/muddy contexts[33]. Our model mathematically predicts the natural emergence of sidewinding in nearly isotropic

environments as a consequence of temporal decoupling between the lateral and vertical undulations, and its selection as advantageous in terms of locomotory performance.

This perspective recently received notable experimental support with evidence of evolutionary convergence in the ventral skin of sidewinding vipers across world deserts[29]. Their skin indeed evolved from the well-documented anisotropic textures characteristic of nonsidewinders to an isotropic one. This, according to our model, maximizes sidewinding locomotion speed (far outperforming other options) and offers a rationale for the observed evolutionary selection (Fig. 3a, b).

**Further bio-physical complexities.** While our model captures broad trends observed in nature, interesting deviations exist. For example, the shovel-nosed snake *Chionactis occipitalis* is well-known to slither on sand[35]. This snake exhibits anisotropic skin texture[29], utilizes a specialized waveform[35], and belongs to the colubrid family whose members are characterized by more slender bodies relative to sidewinding specialists[33]. The combination of waveform and reduced body weight causes the sand to yield and remodel to a lesser degree[35], potentially enabling, in our effective-friction view, an anisotropic response that in turn permits slithering. Further, *C. occipitalis* has been reported to produce $\lambda = 2$ lifting waves[35], which according to our model can generate slithering even in isotropic settings, albeit inefficiently (Fig. 3b). Also interesting is the case of snakes ascending sandy slopes, where sidewinding has been suggested to be an actual

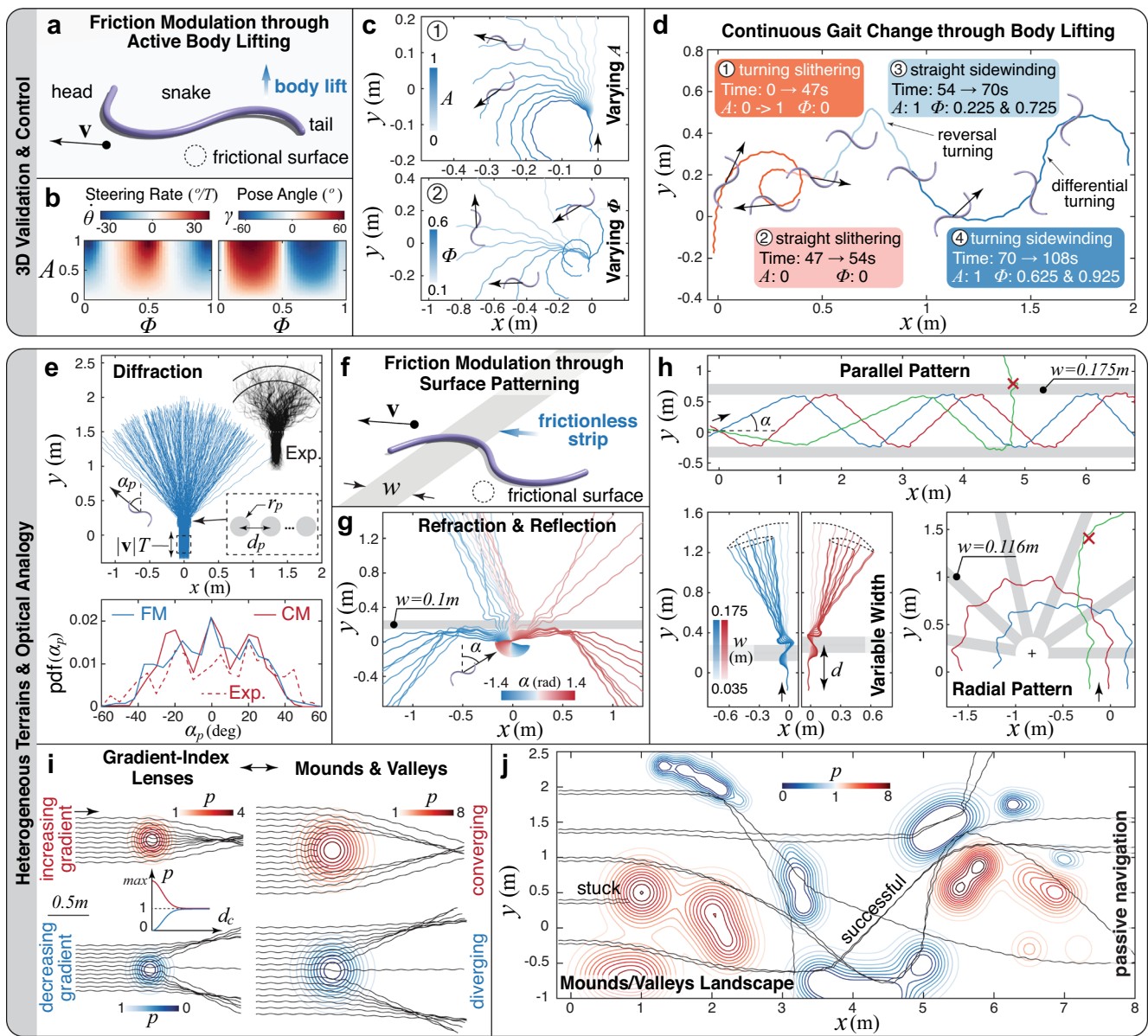

**Fig. 4 Consistency with 3D simulations and control via ground friction design – heterogeneity and optical analogy. a** Schematic of a 3D elastic snake model with internal muscular activation and out-of-plane body lift. **b** Field maps of steering rate and pose angle for $\mu_t/\mu_f = 2$. **c** Trajectories for (1) $\Phi = 0.5$ and $A \in [0, 1]$, (2) $A = 1$ and $\Phi \in [0.1, 0.6]$. We note that $A$ and $\Phi$ here are amplitude and phase offset of the lifting torque wave used to model the snake internal muscular activation (full details in SI). As such they are the dynamic counterparts of the kinematic parameters $A$ and $\Phi$ in the theoretical model. While in both cases the same range and organization of locomotory behaviors is observed, the quantitative values of the two parameter sets match only approximately. **d** Complex trajectories possible by controlling lift amplitude and phase offset. **e** Diffraction pattern and probability density function (pdf) of diffracted angle $\alpha_p$ for simulations of snakes slithering through regularly spaced patches of high friction (friction model — FM, friction is modulated by scaling the local friction coefficients of the patches by a large factor $p$), and comparison with experimental observations of biological snakes traveling through rigid posts (Exp.) and with collision model (CM) simulations[46]. **f** A snake moving on a flat surface ($\mu_t/\mu_f = 10$) patterned with frictionless strips of width $w$. **g** Snakes encountering a frictionless strip are either reflected or refracted depending on incidence angle $\alpha$. **h** Demonstrations of passive trajectory control through friction surface patterning (additional details for all cases in SI). **i** Snakes interacting with heterogeneous ground features of diameter $d_c$ for both increasing and decreasing friction modulation $p$. **j** Snakes passively meander through an heterogeneous frictional contour map. All snakes here utilize a lateral muscular activation function (torque wave) that produces planar gaits of waveform $\kappa$.

necessity to cope with the substrate's high propensity to deform[27,32], rather than an advantageous solution as predicted by our model. Both these examples underscore the role of substrate remodeling, reflow, compaction, and associated resistive forces, further enriching limbless locomotion dynamics. A connection between the granular media resistive forces[31,35–38] and our effective-friction framework might further harmonize theory and observations.

**Consistency with 3D simulations**. Next, we verify the consistency of our findings in 3D direct numerical simulations, whereby a limbless active body is represented as a Cosserat rod (Fig. 4a) equipped with internal muscular activity and interacting with the substrate through contact and friction (Methods/SI). We employ here our Cosserat-based solver *Elastica*[39,40], which has been demonstrated in a number of engineering and biophysical contexts, from the design of bio-hybrid soft robots[41,42] to the

modeling of biological architectures, including snakes[17,39], feathered wings[39] and octopus arms[43,44].

We then instantiate a snake consistent with[13,14], and replace friction modulation waves with torque waves (of the same form) to produce actual body lift (Supplementary Movie 2). As can be seen in Fig. 4b, the 3D model produces variations in steering rates $\dot{\theta}$ and poses $\gamma$ consistent with Fig. 2c, recovering all modes of locomotion in Fig. 2d. Examples of slithering and sidewinding ($\lambda = 1$) are reported in Fig. 4c, showing degrees of turning in line with Fig. 2d. Further informed by Fig. 2d, a repertoire of linear displacements, wide/tight turns, and reversals at varying body poses are concatenated in Fig. 4d (Supplementary Movie 3), illustrating trajectory control in the spirit of[26].

**Control via ground friction design—heterogeneity and optical analogy.** With consistent direct numerical simulations in hand, we switch from a perspective where the snake actively modulates friction to one where locomotory output is instead passively altered by friction patterns on the substrate. The goal is to understand how far our effective–friction perspective can be pushed to investigate heterogeneous environments comprised of small and large-scale 3D features.

We start by considering a recent study proposing an intriguing optical analogy, whereby a snake's body undulations and center of mass represent, respectively, the "wave" and "particle" nature of light[45,46]. There, the authors engineer an environment made of seven rigid cylindrical posts aligned with fixed spacing (Fig. 4e) and let both the biological[46] and robotic[45] snakes slither through the posts, propelled by an approximately planar, stereotypical gait. Surprisingly, the snake-post physical interaction is found to lead to characteristic diffraction patterns. We challenge our approach to reproduce this experiment by taking the drastic step of representing the rigid posts as circular patches of high friction on the ground. As seen in Fig. 4e, our simulations quantitatively match observed deflection distributions, showing how environmental heterogeneities can be successfully modeled as planar friction patterns, simplifying treatment.

Motivated by these results, we further explore the connection with optics, to build intuitive understanding of heterogeneous environments, design passive control strategies, or anticipate failure modes in robotic applications.

As illustrated in Fig. 4f–h, a variety of optical effects can be qualitatively reproduced. For example, by patterning a thin, low-friction strip, we can form an interface, recovering refraction and reflection patterns typical of light transport across two media. Further, we can use this insight and modify the width and spatial arrangement of low-friction strips, to control trajectory deflections, produce U-turns, or even guide snakes along a "channel," analogous to optic fiber light transport (Supplementary Movie 4, 5).

Similarly, we can imitate light convergence/divergence in gradient-index lenses[47] by simply creating friction gradients as illustrated in Fig. 4i. This approach informs the modeling of large-scale (several body lengths) 3D landscape features, such as mounds and valleys. Indeed, slopes may be seen as unbalancing lateral frictional responses, causing the snake to coast in converging (valleys) or diverging (mounds) fashion. To illustrate this concept and further demonstrate the potential for passive, robust control through friction design, we create a topographic map (Fig. 4j) and challenge snakes initialized at different locations to slither through the map, without altering their gaits. As can be seen, snakes meander through the landscape with about ~50% of them making it to the other end, with no active control, showing how friction naturally mediates passive adaptivity to deal with heterogeneities in the environment (Supplementary Movie 6).

In Fig. 4j we also highlight the case of a snake stuck due to high friction. This failure mode exposes the limits of what is essentially an open-loop control strategy based on passive physics. Without active response to sensory feedbacks, there is a natural ceiling to the level of intelligence passive adaptation can offer, particularly under the interference of unforeseen external factors. In this context, our modeling approach and optical analogies provide the understanding and opportunity to devise and experiment with forms of anticipatory[48] or hierarchical[49] control. The latter is particularly appealing, as we envision robots in which a light decision-making process is in charge of producing high-level commands, whose detailed execution is partially or entirely delegated to the physics. In the example of the stuck snake, the high-level controller might just issue a lifting wave template command and then let passive mechanics self-organize a transition to sidewinding that frees the snake. Such an approach relieves the controller from taxing low-level coordination tasks, which, in turn, reduce computing requirements in favor of compact, low-power, and inexpensive on-board processing units, while retaining high levels of adaptivity.

In summary, via minimal theoretical modeling and 3D simulations, our study contextualizes a broad set of observations, both in the biological and robotic domain, through a unified framework centered around effective-friction, actively or passively modulated, on a uniform or heterogeneous substrates, induced by 2D or 3D features, naturally encountered or engineered. It provides a mathematical argument supporting the convergent evolution of sidewinding gaits, while reinforcing the analogy between the limbless terrestrial locomotion and optics, demonstrating its utility for passive trajectory control, with potential applications for bio-inspired engineering.

## Methods

**Planar model of friction modulation.** We adopt the approach of Hu et al.[13,14] wherein the centerline of a snake of length $L$ is modeled as an inextensible planar curve $\hat{s} \in [0, L]$. The center of mass position and average orientation of the snake are denoted by $\bar{\hat{\mathbf{x}}}(t)$ and $\bar{\alpha}(t)$, respectively, and the local position and orientation of each point along the snake's centerline is computed via $\hat{\mathbf{x}}(\hat{s}, \hat{t}) = \bar{\hat{\mathbf{x}}}(t) + I[\mathbf{t}(\hat{s}, \hat{t})]$ and $\alpha(\hat{s}, \hat{t}) = \bar{\alpha}(t) + I[\hat{\kappa}(\hat{s}, \hat{t})]$, respectively, where $\mathbf{t}(\hat{s}, \hat{t}) = (\cos \alpha(\hat{s}, \hat{t}), \sin \alpha(\hat{s}, \hat{t}))$ is the local tangent vector, $\hat{\kappa}(\hat{s}, \hat{t})$ is the local curvature and $I[f(\hat{s}, \hat{t})] = \int_0^{\hat{s}} f(\hat{s}', \hat{t}) d\hat{s}' - \frac{1}{L} \int_0^L \int_0^{\hat{s}} f(\hat{s}', \hat{t}) d\hat{s}' d\hat{s}$ is a mean-zero integration function, which expresses the mathematical machinery that allows us to reconstruct snake's local positions/orientations from the center of mass, global orientation and curvature information (Fig. 1c). The dimensionless form of $\hat{\kappa}(\hat{s}, \hat{t})$ is defined in the main text with all simulations employing $\epsilon = 7$ and $k = 1$, as in[13]. Differentiating $\hat{\mathbf{x}}(\hat{s}, \hat{t})$ twice with respect to time yields

$$\hat{\mathbf{x}}_{tt} = \bar{\hat{\mathbf{x}}}_{tt} + I[-(\bar{\alpha}_t + I[\hat{\kappa}_t])^2 \mathbf{t}] + I[(\bar{\alpha}_{tt} + I[\hat{\kappa}_{tt}])\mathbf{n}]. \tag{1}$$

where $\mathbf{n} = (-\sin \alpha, \cos \alpha)$ is the local normal vector. Writing the snake's dynamics as a force balance of internal $\hat{\mathbf{f}}$ and external $\hat{\mathbf{F}}$ forces per unit length yields

$$\rho \hat{\mathbf{x}}_{tt}(\hat{s}, \hat{t}) = \hat{\mathbf{F}}(\hat{s}, \hat{t}) + \hat{\mathbf{f}}(\hat{s}, \hat{t}), \tag{2}$$

where $\rho$ is the line density of the snake. We then scale Eqs. (1) and (2) by $s = \hat{s}/L$ and $t = \hat{t}/\tau$ to non-dimensionalize the system.

External forces stem entirely from frictional effects captured through the Coulomb friction model, with anisotropy characterized by coefficients in the forward ($\mu_f$), backward ($\mu_b$), and transverse ($\mu_t$) directions. Scaling friction forces such that $\mathbf{F} = \hat{\mathbf{F}}/\rho g \mu_f$ allows us to write the friction force as $\mathbf{F}(s, t) = -N(s, t)\boldsymbol{\mu}(s, t)$ with $\boldsymbol{\mu}(s, t) = \frac{\mu_t}{\mu_f}(\mathbf{u} \cdot \mathbf{n})\mathbf{n} + [(H(\mathbf{u} \cdot \mathbf{t}) + \frac{\mu_b}{\mu_f}(1 - H(\mathbf{u} \cdot \mathbf{t}))](\mathbf{u} \cdot \mathbf{t})\mathbf{t}$, where $\mathbf{u}(s) = \mathbf{x}_t(s)/|\mathbf{x}_t(s)|$ is the unit vector associated with the snake's local velocity direction, and $H = 1/2(1 + \mathrm{sgn}(x))$ is the Heaviside step function used here to distinguish between forward and backward friction components. Here, $\mu_b/\mu_f = 1.5$, in keeping with experimental observations[13]. Previous work[14] and our preliminary investigations found that static friction effects do not appreciably influence the snake's steady-state behavior, thus we did not consider them here. Moreover, we write the friction modulation wave as $N(s, t) = \eta \hat{N}(s, t)$, where $\eta = 1/\int_0^1 \hat{N}(s, t) ds$ is the normalization constant to conserve the overall weight of the snake. Finally, we set Fr = 0.1 throughout, consistent with snakes' typically low values and without lack of generality.

Assuming the total non-dimensionalized internal forces and torques to be zero ($\int_0^1 \mathbf{f}\,ds = 0$ and $\int_0^1 (\mathbf{x} - \overline{\mathbf{x}}) \times \mathbf{f}\,ds = 0$) yields the snake's equation of motion

$$\mathrm{Fr}\,\overline{\mathbf{x}}_{tt}(t) = \int_0^1 -N(s,t)\boldsymbol{\mu}(s,t)\,ds \tag{3}$$

$$\mathrm{Fr}\,\overline{\alpha}_{tt}(t) = \frac{1}{J}\int_0^1 -(\mathbf{x} - \overline{\mathbf{x}}) \times N\boldsymbol{\mu}\,ds \\ + \frac{\mathrm{Fr}}{J}\int_0^1 I[\mathbf{n}] \cdot I[\mathbf{t}(\overline{\alpha}_t + I[\kappa_t])^2] - I[\mathbf{t}] \cdot I[\mathbf{t}I[\kappa_{tt}]]ds, \tag{4}$$

where $J = \int_0^1 (\mathbf{x} - \overline{\mathbf{x}})^2\,ds$ is the moment of inertia. These equations can then be solved for a prescribed non-dimensional curvature $\kappa(s,t)$ and friction scaling term $N(s,t)$.

In all cases considered here, Eqs. (3) and (4) are numerically solved over 10 undulation periods to allow transient effects from startup to dissipate and the snake to reach steady-state behavior. The snake's locomotion behavior is then analyzed in terms of the pose angle $\gamma$, steering rate $\dot{\theta}$, and effective speed $|\mathbf{v}_{\mathrm{eff}}|$ which are illustrated in Fig. 2a. At steady state, the first trajectory metric that can be computed is the pose angle, which is the angle between the snake's average orientation $\overline{\mathbf{t}} = (\cos\overline{\alpha},\ \sin\overline{\alpha})$ and its center of mass velocity direction $\overline{\mathbf{u}}$. The average pose angle over one undulation period is defined as $\gamma = \int_{t_0}^{t_1} \arctan2((\overline{\mathbf{t}} \times \overline{\mathbf{u}}) \cdot \mathbf{e}_z,\ \overline{\mathbf{t}} \cdot \overline{\mathbf{u}})\,\mathrm{d}t/\mathcal{T}$ where $\mathcal{T} = \int_{t_0}^{t_1} \mathrm{d}t$ and $\mathbf{e}_z$ is the unit vector of out-of-plane axis. Use of the arctan2 function is required to ensure $\gamma \in (-\pi, \pi]$. Note that $\mathcal{T}$ is for a nondimensionalized time period, so over one undulation, $\mathcal{T} = 1$. Additional trajectory metrics can be computed by considering the snake's center of mass as a particle undergoing planar motion in polar coordinates, $\overline{\mathbf{x}}(t) = (r\cos\theta, r\sin\theta)$, allowing the snake's trajectory to be quantified in terms of its effective velocity $|\mathbf{v}_{\mathrm{eff}}| = \left| \int_{t_1} r\frac{d\theta}{dt}\,\mathbf{u}_\theta\,dt/\mathcal{T} \right|$ and steering rate $\dot{\theta} = \int_{t_0}^{t_1} \frac{d\theta}{dt}\,dt/\mathcal{T}$ (see SI Note 1 for relevant derivations).

For phase space simulations, a simulation grid was defined with 501 equidistant points in both $A \in [-2, 2]$ and $\Phi \in [0, 1]$, leading to 251k simulations for each of the friction ratios considered. Simulations were performed on the Bridges supercomputing cluster at the Pittsburgh Supercomputing Center.

**3D elastic model of snake locomotion.** 3D elastic simulations of snake locomotion were performed in Elastica[17,39,40] using a Cosserat rod snake model with muscular activation, an approach demonstrated in numerous biophysical applications[17,39,41–43,50,51]. For the Cosserat rod model, we mathematically describe a slender rod by a centerline $\overline{\mathbf{x}}(s,t) \in \mathbb{R}^3$ and a rotation matrix $\mathbf{Q}(s,t) = \{\overline{\mathbf{d}}_1, \overline{\mathbf{d}}_2, \overline{\mathbf{d}}_3\}^{-1}$. Leading to a general relation between frames for any vector $\mathbf{v}$: $\mathbf{v} = \mathbf{Q}\overline{\mathbf{v}}$, $\overline{\mathbf{v}} = \mathbf{Q}^T\mathbf{v}$, where $\overline{\mathbf{v}}$ denotes a vector in the lab frame and $\mathbf{v}$ is a vector in the local frame. Here $s \in [0, L_0]$ is the material coordinate of a rod of rest-length $L_0$, $L$ denotes the deformed filament length and $t$ is time. If the rod is unsheared, $\overline{\mathbf{d}}_3$ points along the centerline tangent $\partial_s\overline{\mathbf{x}} = \overline{\mathbf{x}}_s$ while $\overline{\mathbf{d}}_1$ and $\overline{\mathbf{d}}_2$ span the normal–binormal plane. Shearing and extension shift $\overline{\mathbf{d}}_3$ away from $\overline{\mathbf{x}}_s$, which can be quantified with the shear vector $\boldsymbol{\sigma} = \mathbf{Q}(\overline{\mathbf{x}}_s - \overline{\mathbf{d}}_3) = \mathbf{Q}\overline{\mathbf{x}}_s - \mathbf{d}_3$ in the *local* frame. The curvature vector $\boldsymbol{\kappa}$ encodes $\mathbf{Q}$'s rotation rate along the material coordinate $\partial_s\mathbf{d}_j = \boldsymbol{\kappa} \times \mathbf{d}_j$, while the angular velocity $\boldsymbol{\omega}$ is defined by $\partial_t\mathbf{d}_j = \boldsymbol{\omega} \times \mathbf{d}_j$. We also define the velocity of the centerline $\overline{\mathbf{v}} = \partial_t\overline{\mathbf{x}}$ and, in the rest configuration, the bending stiffness matrix $\mathbf{B}$, shearing stiffness matrix $\mathbf{S}$, second area moment of inertia $\mathbf{I}$, cross-sectional area $A$ and mass per unit length $\rho$. Then, the dynamics[17] of a soft slender body is described by:

$$\rho A \cdot \partial_t^2\overline{\mathbf{x}} = \partial_s\left(\frac{\mathbf{Q}^T\mathbf{S}\boldsymbol{\sigma}}{e}\right) + e\overline{\mathbf{f}} \tag{5}$$

$$\frac{\rho\mathbf{I}}{e} \cdot \partial_t\boldsymbol{\omega} = \partial_s\left(\frac{\mathbf{B}\boldsymbol{\kappa}}{e^3}\right) + \frac{\boldsymbol{\kappa} \times \mathbf{B}\boldsymbol{\kappa}}{e^3} + \left(\mathbf{Q}\frac{\overline{\mathbf{x}}_s}{e} \times \mathbf{S}\boldsymbol{\sigma}\right) \\ + \left(\rho\mathbf{I} \cdot \frac{\boldsymbol{\omega}}{e}\right) \times \boldsymbol{\omega} + \frac{\rho\mathbf{I}\boldsymbol{\omega}}{e^2} \cdot \partial_t e + e\mathbf{c} \tag{6}$$

where Eqs. (5), (6)) represents linear and angular momentum balance at every cross section, $e = |\overline{\mathbf{x}}_s|$ is the local stretching factor, and $\overline{\mathbf{f}}$ and $\mathbf{c}$ are the external force and couple line densities, respectively.

The simulated snakes have length $L = 0.35$ m, diameter $d = 7.7$ mm, and uniform density $\rho = 1000$ kg/m³ to match measurements of milk snakes[13]. The Young's modulus of the filament representing the snake body is $E = 1$ MPa[18] and the gravitational acceleration is $g = 9.81$ m/s². Lateral muscular torques are applied to the filament through the term $\mathbf{c}$ in Eq. (6), and are determined so as to recover curvature profiles consistent with the planar snake model. The period of the lateral undulation is 2 seconds and the forward friction ratio is $\mu_f = 0.089$, resulting in $\mathrm{Fr} = 0.1$ for all simulations. Additional lifting muscular torques are enabled to produce the results of Fig. 4b–d, while non-body lifting snakes have only planar muscular activation. The simulation incorporates the same Coulomb friction model as in the planar snake model above. More details on our Cosserat rod model,

discretization parameteres, and additional information regarding the different cases of Fig. 4 are available in the supplementary information.

**On-line, interactive sandbox.** The 2D planar results presented in the paper serve to explore and intuit snake gait adaptations to terrains of various nature. To aid this exploration, we first open-source our computational code under a liberal license (see Code availability). To further enable a seamless research/educational experience and to disencumber scientists/students from the process of installing the essential computational software stack, we provide an interactive sandbox built atop our code. This sandbox is free, open-source, hosted online, and is accessible from any modern web browser running on personal devices from mobile phones to laptops. It can be accessed using the link provided in the Code Availability section. In this sandbox, users can utilize intuitive sliders and drop-down menus to change dynamical parameters, which are then used to run simulations asynchronously before presenting results in an interactive plot.

## Data availability
Raw data from all simulations is available from the authors upon request.

## Code availability
An interactive, free, open-source, online sandbox demonstrating our 2D planar snake gait results is available at https://gazzolalab.github.io/kinematic_snake_sandbox. The numerical simulations powering this sandbox were executed using custom `Python` code, which we also open-source, accessible at https://github.com/GazzolaLab/kinematic_snake. Simulations of the 3D snake were performed using a C++ version of Elastica, our open-source numerical simulator for Cosserat rod dynamics, which is accessible at https://www.cosseratrods.org/.

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

## Acknowledgements

We thank Henry Astley for the useful discussions and careful proof-reading. This study is jointly funded by NSF EFRI C3 SoRo #1830881 (M.G.), NSF CAREER #1846752 (M.G.), and ONR MURI N00014-19-1-2373 (M.G.). We also thank the Blue Waters project (OCI-0725070, ACI-1238993), a joint effort of the University of Illinois at Urbana-Champaign and its National Center for Supercomputing Applications, and the Extreme Science and Engineering Discovery Environment (XSEDE) Bridges, supported by National Science Foundation grant number ACI-1548562, at the Pittsburgh super-computing center through allocation TG-MCB190004.

## Author contributions

X.Z., N.N., T.P., and M.G. designed the research. X.Z., N.N., and T.P. performed the research. X.Z., N.N., T.P., and M.G. analyzed the data. X.Z., N.N., T.P., and M.G. wrote the paper.

## Competing interests
The authors declare no competing interests.
