## [Peer Review File · Nature Communications]

Editorial Note: This manuscript has been previously reviewed at another journal that is not operating a transparent peer review scheme. This document only contains reviewer comments and rebuttal letters for versions considered at Nature Communications

REVIEWERS' COMMENTS

Reviewer #1 (Remarks to the Author):

In this paper, motivated by a possible convergence of terrestrial limbless locomotion strategies ultimately determined by interfacial effects, the authors show how both 3D gait alterations and locomotory adaptations to heterogeneous terrains can be understood through the lens of local friction modulation. The study outlines a unified view that connects active and passive 3D mechanics with heterogeneous interfacial effects to explain a broad set of biological observations, and potentially inspire engineering design. The work of this paper is practical and logical. In general, the manuscript is well organized, and the contributions are convincing. Besides, I have the following suggestions which may help the authors improve the quality of the manuscript.

1. This study outlines a unified view that connects active and passive 3D mechanics with heterogeneous interfacial effects to explain a broad set of biological observations, and potentially inspire engineering design. This is a very interesting and novel point of view. However, the authors need to further discuss the principles and mechanisms of the high environmental adaptability of biological snakes under the interference of other complex factors in the complex terrain, and further supplement the potential connections and differences with the optical principles proposed in this paper.
2. The authors further explore the connection with optics, to build intuitive understanding of heterogeneous environments, design passive control strategies or anticipate failure modes in robotic applications. We can see from the movie and Fig. 4 that when individual snakes passively meandering through the heterogeneous friction contour map, they may be trapped in the local extreme value and cannot be traversed. Therefore, did the authors consider adding some optimizations to further improve the modeling and planning in the passive meandering state?
3. This paper provides a mathematical argument supporting the convergent evolution of sidewinding gaits, while reinforcing the analogy between limbless terrestrial locomotion and optics, demonstrating its utility for passive trajectory control, with potential applications for bio-inspired engineering. This paper mainly considers the sidewinding gaits of snakes for modeling and analysis. Could the authors further discuss the realization of the biological snake-inspired switching movement mode of different gaits in different heterogeneous environments, so as to better combine the bionics and optics fields?

Reviewer #2 (Remarks to the Author):

This manuscript provides a simple yet elegant model for limbless locomotion and tests it through 2D and 3D simulations with active and passive friction modulations. In addition, the study shows that environmental heterogeneities can be modeled as planar friction patterns and can be engineered to allow for passive snake navigation. This is a high-impact, rigorous, and well-crafted manuscript that could be of great interest to the interdisciplinary readership of this journal. my comments are listed in the following:

- 1- I am not sure if emphasizing "everything-is-friction" is appropriate particularly for mobility in granular media. The authors show that when there is no friction anisotropy (e.g., in mud or sand), the

dominant gait is sidewinding. That is mainly because the ground reaction forces in such media can be provided due to the substrate deformation rather than friction. I suggest adding a brief statement to clarify this.

2- Some of the statements about biological snakes that are mentioned when discussing figure 3 may not be accurate (on page 3). For instance, some species of snakes such as sidewinder rattlesnakes prefer to do sidewinding even on hard ground on which there may be significant friction anisotropy. In contrast, some snake species such as the Mojave shovel-nosed snakes can successfully slither on sandy terrains. The authors may consider adding a statement about the complexities of biological systems that may not be captured by the proposed model.

3- The optical analogies presented in Fig. 4g-h may distract the audience from the main message of this study and could potentially be moved to the supplementary material.

Reviewer #3 (Remarks to the Author):

I want to commend the authors on a wonderful paper--likely one of my favorites to read in many years! The simulations are timely and interesting, and should have value in setting up biological hypotheses for control of highly damped terrestrial locomotion (much more common than realized as the authors point out) and should aid roboticists in designing new limbless locomotors that can modulate surface forces to effect locomotion in complex terrain. Of course I could quibble and state that a frictional model of diverse terrain might be a bit simplistic, but I will answer that quibble by noting that it seems to be a good starting point. Another issue I might have is that this work seems to be on flat "ground" and (in sidewinding at least) the ascent of slopes is the challenge that seems to really require sidewinding--e.g. in the Marvi et al 2014 paper some snakes can move on sand via lateral undulation but this becomes harder and harder as slope increases. I believe this point is discussed in the Astley 2020 JEB review. In granular media, it is the reflow of the material that can lead to interesting challenges in limbless (e.g. see Schiebel et al, *elife* 2020) and limbed (e.g. see Shrivastava et al, *Science Robotics*, 2020 or Mazouchova et al, *B&B* 2013).

Otherwise, I would advocate that this paper be published without delay (provided the authors address the above)--exciting and cool work!

And PS, I love the optics analogies!

Response to Reviewer 1:

Friction modulation in limbless, three-dimensional gaits and heterogeneous terrains

Xiaotian Zhang, Noel Naughton, Tejaswin Parthasarathy, and Mattia Gazzola

We thank the reviewer for her/his valuable time, consideration, and positive assessment. In the following, the comments of the reviewer are listed, followed by our responses. All modifications to the manuscript are highlighted in red in the manuscript and are also reported below, for the reviewer's convenience.

We hope that the reviewer considers our answers acceptable, and the revised manuscript suitable for publication.

In this paper, motivated by a possible convergence of terrestrial limbless locomotion strategies ultimately determined by interfacial effects, the authors show how both 3D gait alterations and locomotory adaptations to heterogeneous terrains can be understood through the lens of local friction modulation. The study outlines a unified view that connects active and passive 3D mechanics with heterogeneous interfacial effects to explain a broad set of biological observations, and potentially inspire engineering design. The work of this paper is practical and logical. In general, the manuscript is well organized, and the contributions are convincing. Besides, I have the following suggestions which may help the authors improve the quality of the manuscript.

We thank the reviewer for her/his positive assessment of the manuscript. Below we address the reviewer's suggestions, which we have incorporated into the manuscript and believe they have helped clarify and improve the work as a whole.

1. This study outlines a unified view that connects active and passive 3D mechanics with heterogeneous interfacial effects to explain a broad set of biological observations, and potentially inspire engineering design. This is a very interesting and novel point of view. However, the authors need to further discuss the principles and mechanisms of the high environmental adaptability of biological snakes under the interference of other complex factors in the complex terrain, and further supplement the potential connections and differences with the optical principles proposed in this paper.

2. The authors further explore the connection with optics, to build intuitive understanding of heterogeneous environments, design passive control strategies or anticipate failure modes in robotic applications. We can see from the movie and Fig. 4 that when individual snakes passively meandering through the heterogeneous friction contour map, they may be trapped in the local extreme value and cannot be traversed. Therefore, did the authors consider adding some optimizations to further improve the modeling and planning in the passive meandering state?

3. This paper provides a mathematical argument supporting the convergent evolution of sidewinding gaits, while reinforcing the analogy between limbless terrestrial locomotion and optics, demonstrating its utility for passive trajectory control, with potential applications for bio-inspired engineering. This paper mainly considers the sidewinding gaits of snakes for modeling and analysis. Could the authors further discuss the realization of the biological snake-inspired switching movement mode of different gaits in different heterogeneous environments, so as to better combine the bionics and optics fields?

We thank the reviewer for the above suggestions, which indeed prompted us to clarify important aspects of our work, and to introduce an additional online, interactive simulator (see last comment of this document) to explore some of the concepts presented in this paper. Since the three points raised by the reviewer touch upon various aspects of gait generation, control and adaptivity, we address them all in the main text through a new and dedicated discussion, using as a starting point the failure example of the stuck snake, highlighted by the reviewer.

In general, the topic of principles and mechanisms of environmental adaptability in biological snakes is vast and complex and has challenged researchers for decades. While many biological specializations –depending on snakes' habitats, habits and sensing capabilities— have been

observed and reported, mechanistic explanations are missing in all but the simplest cases. A quote from Prof. Bruce Jayne's recent review on snake locomotion (Jayne, "What Defines Different Modes of Snake Locomotion", *Integrative and Comparative Biology*, 2020) well summarizes this point: "...Furthermore, unlike the complex structural variation that snakes often encounter naturally, most laboratory studies have understandably focused on using simpler conditions to elicit different modes of snake locomotion and studying movement mainly in a horizontal plane."

The same point is remarked in Schiebel et al, "Mechanical diffraction reveals the role of passive dynamics in a slithering snake", *PNAS*, 2019: "...understanding movement in heterogeneous terrain remains a frontier in locomotion studies."

Since so little is understood about limbless locomotion on heterogenous substrates, and since this literature is already reviewed (and in fact quantitatively modelled) in our manuscript, we refrain from providing an additional, broader (and forcibly speculative) discussion. Nonetheless, Point 1 raised by the reviewer provides us with the opportunity to better contextualize our novel understanding of passive, friction-mediated control, within the scope of robotic control integration.

We start our discussion (Page 6) by underscoring the case of the stuck snake (Point 2 raised by the reviewer) and use this failure mode (which is now also clearly highlighted in Fig 4j) to introduce the limits of a purely passive control strategy based on frictional environment design. Such an approach is, in fact, essentially a form of open loop control, and as such it presents inherent limitations to the level of intelligence or adaptivity that can be achieved. Therefore, to deal with being stuck or, more generally, to cope with the interference of unforeseen external factors and complexities (Point 1 raised by the reviewer), a robotic snake would need some form of active control informed by sensory feedbacks (as is the case in biological snakes).

Still, the passive organization of trajectories and stereotypical locomotory behaviors enabled by our analysis provides the understanding and opportunity to outsource taxing low-level coordination tasks to the physics, thus simplifying control overall. Then, our modeling approach and optical analogies can be thought of as a testing ground to develop forms of anticipatory or hierarchical control, with the goal of minimizing algorithmic complexity and computing requirements. In this context, we find particularly appealing the hierarchical approach, as we envision robots in which a light decision-making process is in charge of producing high-level commands, whose detailed execution is partially or entirely delegated to the physics. In the example of the stuck snake, the high-level controller might just issue a lifting wave template command and then let passive mechanics self-organize a transition to sidewinding that frees the snake. Such approach reduces computing needs in favor of compact, low-power, and inexpensive on-board processing units, while retaining high levels of adaptivity.

This is now encapsulated in the last paragraph before conclusions (Page 6), reported here for the reviewer's convenience:

"In Fig. 4j we also highlight the case of a snake stuck due to high friction. This failure mode exposes the limits of what is essentially an open-loop control strategy based on passive physics. Without active response to sensory feedbacks there is a natural ceiling to the level of intelligence passive adaptation can offer, particularly under the interference of unforeseen external factors. In this context, our modeling approach and optical analogies provide the understanding and opportunity to devise and experiment with forms of anticipatory [49] or hierarchical [50] control."

The latter is particularly appealing, as we envision robots in which a light decision-making process is in charge of producing high-level commands, whose detailed execution is partially or entirely delegated to the physics. In the example of the stuck snake, the high-level controller might just issue a lifting wave template command and then let passive mechanics self-organize a transition to sidewinding that frees the snake. Such an approach relieves the controller from taxing low-level coordination tasks, which, in turn, reduce computing requirements in favor of compact, low-power, and inexpensive on-board processing units, while retaining high levels of adaptivity.”

We believe that this discussion addresses also the second point raised by the reviewer, who asks whether we considered adding optimizations to improve modelling and planning in the meandering study. Our proposed hierarchical control approach goes in the direction of improving navigation, through a combination of active high-level control —which may well entail a planning algorithm to chart an approximate trajectory— and passive, friction-based low-level control for unsupervised execution and/or local correction.

Relative instead to potential improvements in the modeling of heterogenous features (which links back to Point 1: “..other complex factors in the complex terrain..”), we have added a new section in Page 4-5 titled “Further bio-physical complexities”. In this new section we discuss observations of snake behaviors that interestingly deviate from our model predictions, underscoring the role of granular substrate’s deformation, remodeling or reflow. In order to better reflect these phenomena, a possible avenue of future research is proposed, to establish a connection between resistive force methods for granular media and our friction-based framework.

The new section (Page4-5) is reported here for the reviewer’s convenience:

*“Further bio-physical complexities. While our model captures broad trends observed in nature, interesting deviations exist. For example, the shovel-nosed snake *Chionactis occipitalis* is well-known to slither on sand [37]. This snake exhibits anisotropic skin texture [29], utilizes a specialized waveform [37], and belongs to the colubrid family whose members are characterized by more slender bodies relative to sidewinding specialists [32]. The combination of waveform and reduced body weight causes the sand to yield and remodel to a lesser degree [37], potentially enabling, in our effective–friction view, an anisotropic response that in turn permits slithering. Further, *C. occipitalis* has been reported to produce $\lambda = 2$ lifting waves [37], which according to our model can generate slithering even in isotropic settings, albeit inefficiently (Fig. 3b). Also interesting is the case of snakes ascending sandy slopes, where sidewinding has been suggested to be an actual necessity to cope with the substrate’s high propensity to deform [27, 35], rather than an advantageous solution as predicted by our model. Both these examples underscore the role of substrate remodeling, reflow, compaction, and associated resistive forces, further enriching limbless locomotion dynamics. A connection between granular media resistive forces [34, 37–40] and our effective–friction framework might further harmonize theory and observations.”*

Relative to Point 3 raised by the reviewer, we interpreted it as a request to elucidate how switching between locomotory behaviors would be practically implemented in a bionic control setting.

We start by noting that in our analysis forward sidewinding, turning sidewinding, forward slithering, turning slithering, backward slithering and spinning are all locomotory outputs that emerge from the same gait template (lateral + lifting wave). Assuming that the lateral waveform is maintained unchanged, the manifestation of each one of the above modes is governed by three parameters only, lifting wave amplitude A , offset Φ_i , (which together represent gait, i.e. body deformations)

and skin texture-substrate friction ratio μ_t/μ_f (which captures the environmental interaction). It is also perhaps useful to note that a given gait (A, Φ) might produce sidewinding in a certain environment (most notably isotropic ones such as deserts), while instead producing slithering on a different substrate, such as a hard rock. Thus, as a snake moves from one environment to another with perfectly unaltered gait, a transition or switch between locomotory behaviors automatically takes place in a self-organized fashion. Thus, *“the realization of the biological snake-inspired switching movement mode of different gaits in different heterogeneous environments”* pointed out by the reviewer boils down to either do nothing (the same gait manifests automatically in different locomotory behaviors depending on the local frictional context), or modify A or Φ according to the phase-spaces of Fig. 3. This provides a minimal, simple, and appealing control mechanism to be taken advantage of in a bionic implementation.

Relative to the optic analogy, our message is that we may even entirely remove (A, Φ) and pre-program the snake’s behavior and trajectory by locally manipulating μ_t/μ_f through the design of frictional surface patterns. With in mind potential robotic applications, our optical analogies are useful in a few ways: 1) Provide an intuition about the snake behavior in relation to substrate heterogenous features, and inform us about potential failure modes we will need to anticipate. 2) Simplify the general treatment of complex heterogenous environments (which is a long-standing challenge), allowing us to computationally experiment with various forms of active + passive control. 3) Enable fully passive ways of controlling and distributing robots across a well-characterized domain. We might imagine, for example, patterning a warehouse floor to passively deploy robots in a desired spatial distribution to target locations, or vice-versa collect them from various locations, without no control nor gait alterations of sort.

We believe that the new additional discussions of Page 6 (on control) and of Page 4-5 (on potential avenues for modeling improvement), in combination with the already existing discussion on transitions among locomotory behaviors relative to Fig 3 and Fig 4, should convey the points above illustrated, incorporating the reviewer’s suggestions, albeit in a concise form.

We conclude by pointing the reviewer to an additional resource that now complements the manuscript, and that was inspired precisely by the reviewer’s comments. We indeed have created an online, freely-accessible and intuitive sandbox for readers and users to experiment with different actuation and friction conditions and observe in real time resulting locomotory outputs. We believe this could be an impactful tool for researchers, for the simply curious reader and for outreach activities.

Our sandbox is live and can be found at:

https://gazzolalab.github.io/kinematic_snake_sandbox/snake_sandbox.html

The sandbox will be also progressively expanded in time, to encompass more complex and advanced features. A brief description of this tool can now be found in the Methods section of the manuscript.

Finally, we wish to thank the reviewer for her/his careful evaluation, critical comments and suggestions. We believe they helped us improve our manuscript as a whole, potentially broadening its appeal across readerships. We hope the reviewer finds our answers satisfactory and the manuscript suitable for publication.

References used in this response:

- B.C. Jayne. *Integrative and comparative biology* 60, 1 (2020).
- P.E. Schiebel, J. M. Rieser, A. M. Hubbard, L. Chen, D. Z. Rocklin, and D. I. Goldman. *PNAS* 116, 11 (2019).

Response to Reviewer 2:

Friction modulation in limbless, three-dimensional gaits and heterogeneous terrains

Xiaotian Zhang, Noel Naughton, Tejaswin Parthasarathy, and Mattia Gazzola

We thank the reviewer for her/his valuable time, consideration, and positive assessment. In the following, the comments of the reviewer are listed, followed by our responses. All modifications to the manuscript are highlighted in red in the manuscript and are also reported below, for the reviewer's convenience.

We hope that the reviewer considers our answers acceptable, and the revised manuscript suitable for publication.

This manuscript provides a simple yet elegant model for limbless locomotion and tests it through 2D and 3D simulations with active and passive friction modulations. In addition, the study shows that environmental heterogeneities can be modeled as planar friction patterns and can be engineered to allow for passive snake navigation. This is a high-impact, rigorous, and well-crafted manuscript that could be of great interest to the interdisciplinary readership of this journal. my comments are listed in the following:

We thank the reviewer for her/his positive assessment of the manuscript. Below we address the reviewer's suggestions, which we have incorporated into the manuscript and believe they have helped clarify, nuance, and improve the work as a whole.

1- I am not sure if emphasizing "everything-is-friction" is appropriate particularly for mobility in granular media. The authors show that when there is no friction anisotropy (e.g., in mud or sand), the dominant gait is sidewinding. That is mainly because the ground reaction forces in such media can be provided due to the substrate deformation rather than friction. I suggest adding a brief statement to clarify this.

We agree with the reviewer. Indeed, the *'everything-is-friction'* terminology might suggest that the snake's behaviors modeled in this work are a consequence of friction forces alone, which is not necessarily the case. Therefore, we replaced throughout the text the expression *'everything-is-friction approach'* with *'effective—friction approach'*. The intent is to remove the above ambiguity and clarify that we do not necessarily attribute the observed snakes' behaviors strictly to friction per se, but rather we capture and 'flatten' complex interfacial dynamics, including granular substrate deformations and remodeling, onto a planar ground through an *effective* friction (or friction ratio).

At the end of the introduction (Page 1) we added:

"Here, motivated by a possible evolutionary convergence of limbless movements ultimately determined by interfacial effects, the roles of both 3D body deformations and environmental heterogeneities are connected through and modeled as planar friction modulations. Thus, by homogenizing the complex interaction between limbless creatures and substrate features into a spatially and temporally varying 2D frictional field, we combine theory and simulations to establish an effective—friction perspective that coherently explains a broad set of observations."

This also made explicit as we discuss the meaning of the modulation function $N(s,t)$ in the model section (Page 3):

"Finally, the function $N(s, t) = \eta \hat{N}(s, t)$, with η a normalization factor, models body lift as local weight redistributions, leading to effective—friction modulations along the snake. This modulation implicitly gives rise to a temporally and spatially varying field of frictional force magnitudes, actively controlled by the snake."

Further, while discussing our results for isotropic friction ratios (Page 3), we now provide a connection between effective friction approximation and substrate deformation. Page 3, third paragraph:

*“However, sidewinding is found to be significantly faster (Fig 3b), thus proving advantageous in environments such as sandy deserts or mudflats, characterized by low **effective-friction** ratios on account of their propensity to yield under stress [34, 35].”*

We believe that this effective-friction abstraction is key, affording our reduced-order model approach broad explanatory power. Nonetheless, it is also a simplification and it does not allow neatly separating or dissecting the various roles of friction and, for example, as pointed out by the reviewer, additional granular media effects. We then created a new section titled *“Further bio-physical complexities”*, in which we now address deviations from and limitations of our modeling approach, and potential for improvement. In there, we discuss the case of the shovel-nosed snake (raised by the reviewer as well – see next answer) and further emphasize the role of granular effects in enriching limbless locomotion dynamics.

The new section (Page 4) reads:

*“**Further bio-physical complexities.** While our model captures broad trends observed in nature, interesting deviations exist. For example, the shovel-nosed snake *Chionactis occipitalis* is well-known to slither on sand [37]. This snake exhibits anisotropic skin texture [29], utilizes a specialized waveform [37], and belongs to the colubrid family whose members are characterized by more slender bodies relative to sidewinding specialists [32]. The combination of waveform and reduced body weight causes the sand to yield and remodel to a lesser degree [37], potentially enabling, in our effective–friction view, an anisotropic response that in turn permits slithering. Further, *C. occipitalis* has been reported to produce $\lambda = 2$ lifting waves [37], which according to our model can generate slithering even in isotropic settings, albeit inefficiently (Fig. 3b). Also interesting is the case of snakes ascending sandy slopes, where sidewinding has been suggested to be an actual necessity to cope with the substrate’s high propensity to deform [27, 35], rather than an advantageous solution as predicted by our model. Both these examples underscore the role of substrate remodeling, reflow, compaction, and associated resistive forces, further enriching limbless locomotion dynamics. A connection between granular media resistive forces [34, 37–40] and our effective–friction framework might further harmonize theory and observations.”*

2- Some of the statements about biological snakes that are mentioned when discussing figure 3 may not be accurate (on page 3). For instance, some species of snakes such as sidewinder rattlesnakes prefer to do sidewinding even on hard ground on which there may be significant friction anisotropy. In contrast, some snake species such as the Mojave shovel-nosed snakes can successfully slither on sandy terrains. The authors may consider adding a statement about the complexities of biological systems that may not be captured by the proposed model.

The reviewer points out two important observations: 1) the fact that some species of snakes, most notably the sidewinder rattlesnake *Crotalus cerastes*, sidewind beyond sandy environments where friction isotropy might not be well approximated; 2) the case of shovel-nosed snakes that possess demonstrated abilities to slither on sand.

We address them separately.

First, we now explicitly mention the sidewinder rattlesnake as an example of snakes that do sidewind outside sandy environments. We also point out that this behavior is still relatively rare (as reported in *Tingle, Integrative and Comparative Biology, 2020*) and, importantly, not

necessarily in contrast with our predictions. Indeed, as can be seen in Fig 3, sidewinding can exist beyond perfect isotropy. For example, for a friction ratio = 2 sidewinding can be attained (blue regions in the corresponding phase-space plot of panel a) and is competitive with slithering in terms of speeds (Fig 3b). The potential downside, according to our model, is that in anisotropic frictional environments sidewinding can be achieved at the price of higher liftings, which may be inconvenient or energetically unfavorable, thus providing a mechanistic explanation for its less widespread use outside sandy or muddy terrains.

We now briefly discuss these points in Page 4, first paragraph

*“This is consistent with the fact that sidewinders and slitherers are comparably fast in their respective habitats [14]. It is also consistent with observations that sidewinding rarely occurs outside of sandy and muddy terrains [32], although some desert sidewinding-specialists, most notably *Crotalus cerastes*, do sidewind on substrates other than isotropic sand [32]. This is not in contrast with our model, which, in fact, allows for sidewinding at moderate friction ratios, albeit at the cost of increased body lifts (Fig. 3a).”*

Second, we discuss the case of the shovel-nosed snake.

We created the new section “*Further bio-physical complexities*” (reported in full in the above answer) in which we address deviations and limitations of our modeling approach, and potential for improvement. In there, we discuss the case of the shovel-nosed snake (as well as the case of ascending sandy slopes, in which snakes find it harder and harder to climb without sidewinding, as the slope increases – a point raised by another reviewer).

Relative to the shovel-nosed and ascending cases, we believe our model might hint to a potential, partial explanation (or at the very least, that it is not in direct contrast to observations). We see two ways, potentially connected, to look at the problem. First, the shovel-nosed has been reported to produce $\lambda = 2$ waves (Schiebel et al, elife 2020), which we predict can generate locomotion on sand (Fig 3b), albeit at the cost of reduced forward speeds. Second, the shovel-nosed snake also presents distinctive aspects that might contribute to explain its behavior. As mentioned in the “*Further bio-physical complexities*” above, the shovel-nosed snake exhibits anisotropic skin texture, utilizes an actuation waveform different from sidewinding vipers, and belongs to the colubrid family that is characterized by more slender bodies. The combination of waveform and reduced body weight has been observed to cause sand to yield to a lesser degree (Schiebel et al, elife 2020). This plausibly enables an anisotropic effective-friction response, that in turn permits slithering (Fig 3). This second aspect connects back to the first point raised by the reviewer, that indeed granular substrate deformation, reflow and remodeling are enriching aspects of limbless terrestrial locomotion which are not treated in detail in our model, but rather ‘lumped’ in an overall effective frictional response. We do emphasize this aspect the “*Further bio-physical complexities*” section, and point to it as an important avenue of future work.

As mentioned, all of these points can now be found in the *Further bio-physical complexities* (Page 4 of the manuscript main text, and Page 3, Answer 1 in this document).

3- The optical analogies presented in Fig. 4g-h may distract the audience from the main message of this study and could potentially be moved to the supplementary material.

While we understand that the substrate design and optical analogy portion of the manuscript might distract from the biological component, we feel that it contributes to the paper as a whole in a few ways.

First, it is an indirect confirmation of our effective—friction view. Indeed, if we can think of 3D snake deformations, at their very core, as means of modulating friction so that locomotory outputs can be recovered through a planar model coupled to locally variable frictional effects, then we should be able to reverse this view and control snake locomotion by directly engineering friction patterns on the ground.

Second, they do provide a way to gain intuition into ground heterogeneity effects, which is a challenging topic from theoretical, computational, and experimental perspectives. Optical analogies then can be thought of as an additional resource in the community's toolbox to begin to address these aspects, both in the biological and engineering domain.

Third, this brings us to our third main reason to include this analysis, which is its potential interest in engineering, robotics, and control. To this end we have expanded the discussion at the end of Page 6.

Page 6, last paragraphs before Conclusions

“In Fig. 4j we also highlight the case of a snake stuck due to high friction. This failure mode exposes the limits of what is essentially an open-loop control strategy based on passive physics. Without active response to sensory feedbacks there is a natural ceiling to the level of intelligence passive adaptation can offer, particularly under the interference of unforeseen external factors. In this context, our modeling approach and optical analogies provide the understanding and opportunity to devise and experiment with forms of anticipatory [49] or hierarchical [50] control. The latter is particularly appealing, as we envision robots in which a light decision-making process is in charge of producing high-level commands, whose detailed execution is partially or entirely delegated to the physics. In the example of the stuck snake, the high-level controller might just issue a lifting wave template command and then let passive mechanics self-organize a transition to sidewinding that frees the snake. Such an approach relieves the controller from taxing low-level coordination tasks, which, in turn, reduce computing requirements in favor of compact, low-power, and inexpensive on-board processing units, while retaining high levels of adaptivity.”

In light of these considerations, we respectfully opt for retaining the optical analogy discussion in the main text.

Finally, we felt compelled to add an additional resource complementing our manuscript: an online, freely-accessible and intuitive sandbox for readers and users to experiment with different actuation and friction conditions and observe in real time resulting locomotory outputs. We believe this could be an impactful tool for researchers, for the simply curious reader and for outreach activities.

Our sandbox is live and can be found at:

https://gazzolalab.github.io/kinematic_snake_sandbox/snake_sandbox.html

The sandbox will be also progressively expanded in time, to encompass more complex and advanced features.

A brief description of this tool can now be found in the Methods section of the manuscript.

Finally, we wish to thank the reviewer for her/his careful evaluation, critical comments and suggestions. We believe they helped us improve our manuscript as a whole, potentially broadening its appeal across readerships. We hope the reviewer finds our answers satisfactory and the manuscript suitable for publication.

References used in this response:

- J. L. Tingle, *Integrative and Comparative Biology* 60, 1 (2020).
- J. M. Rieser, J. L. Tingle, D. I. Goldman, J. R. Mendelson, et al., *PNAS* 118, 6 (2021).
- P. E. Schiebel, H. C. Astley, J. M. Rieser, S. Agarwal, C. Hubicki, A. M. Hubbard, K. Diaz, J. R. Mendelson III, K. Kamrin, and D. I. Goldman, *Elife* 9, e51412 (2020).

Response to Reviewer 3:

Friction modulation in limbless, three-dimensional gaits and heterogeneous terrains

Xiaotian Zhang, Noel Naughton, Tejaswin Parthasarathy, and Mattia Gazzola

We thank the reviewer for her/his valuable time, consideration, and positive assessment. In the following, the comments of the reviewer are listed, followed by our responses. All modifications to the manuscript are highlighted in red in the manuscript and are also reported below, for the reviewer's convenience.

We hope that the reviewer considers our answers acceptable, and the revised manuscript suitable for publication.

I want to commend the authors on a wonderful paper--likely one of my favorites to read in many years! The simulations are timely and interesting, and should have value in setting up biological hypotheses for control of highly damped terrestrial locomotion (much more common than realized as the authors point out) and should aid roboticists in designing new limbless locomotors that can modulate surface forces to effect locomotion in complex terrain.

We thank the reviewer for her/his very positive assessment of the manuscript. Below we address the reviewer's suggestions, which we have incorporated into the manuscript and believe they have helped clarify and improve the work as a whole.

Of course I could quibble and state that a frictional model of diverse terrain might be a bit simplistic, but I will answer that quibble by noting that it seems to be a good starting point. Another issue I might have is that this work seems to be on flat "ground" and (in sidwinding at least) the ascent of slopes is the challenge that seems to really require sidwinding--e.g. in the Marvi et al 2014 paper some snakes can move on sand via lateral undulation but this becomes harder and harder as slope increases. I believe this point is discussed in the Astley 2020 JEB review. In granular media, it is the reflow of the material that can lead to interesting challenges in limbless (e.g. see Schiebel et al, elife 2020) and limbed (e.g. see Shrivastava et al, Science Robotics, 2020 or Mazouchova et al, B&B 2013).

We agree with the reviewer that this reduced-order modeling approach is a simplified yet useful starting point for elucidating broad principles governing limbless locomotion. Additionally, we believe that future work can expand this perspective to incorporate additional complexities, particularly granular media effects.

In the revised text we now explicitly address these points. First, we replaced throughout text the expression '*everything-is-friction approach*' with '*effective--friction approach*'. The intent is to remove ambiguity and clarify that we do not necessarily attribute the observed snakes' behaviors strictly to friction per se, but rather we capture and 'flatten' complex interfacial dynamics, including granular media effects, onto a planar ground through an effective friction (or friction ratio).

At the end of the introduction (Page 1) we added:

"Here, motivated by a possible evolutionary convergence of limbless movements ultimately determined by interfacial effects, the roles of both 3D body deformations and environmental heterogeneities are connected through and modeled as planar friction modulations. Thus, by homogenizing the complex interaction between limbless creatures and substrate features into a spatially and temporally varying 2D frictional field, we combine theory and simulations to establish an effective--friction perspective that coherently explains a broad set of observations."

This also made explicit as we discuss the meaning of the modulation function $N(s,t)$ in the model section (Page 3):

"Finally, the function $N(s, t) = \eta \hat{N}(s, t)$, with η a normalization factor, models body lift as local weight redistributions, leading to effective--friction modulations along the snake. This modulation implicitly gives rise to a temporally and spatially varying field of frictional force magnitudes, actively controlled by the snake."

Further, while discussing our results for isotropic friction ratios (Page 3), we now provide a connection between effective friction approximation and substrate deformation. Page 3, third paragraph:

*“However, sidewinding is found to be significantly faster (Fig 3b), thus proving advantageous in environments such as sandy deserts or mudflats, characterized by low **effective-friction** ratios on account of their propensity to yield under stress [34, 35].”*

We believe that this effective-friction abstraction is key, affording our reduced-order model approach broad explanatory power. Nonetheless, as pointed out by the reviewer, it is also a simplification and it does not allow neatly separating or dissecting the role of friction and, for example, additional granular media effects. We then created a new section titled “*Further bio-physical complexities*”, in which we now address deviations from and limitations of our modeling approach, and potential for improvement. In there, we discuss the case of the shovel-nosed snake (raised by another reviewer) as well as ascent of slopes, and further emphasize the role of granular effects in enriching locomotion dynamics. We also emphasize a connection between granular media resistive forces and our effective-friction framework as an important avenue of future work.

The new section (Page 4) reads:

*“**Further bio-physical complexities.** While our model captures broad trends observed in nature, interesting deviations exist. For example, the shovel-nosed snake *Chionactis occipitalis* is well-known to slither on sand [37]. This snake exhibits anisotropic skin texture [29], utilizes a specialized waveform [37], and belongs to the colubrid family whose members are characterized by more slender bodies relative to sidewinding specialists [32]. The combination of waveform and reduced body weight causes the sand to yield and remodel to a lesser degree [37], potentially enabling, in our effective–friction view, an anisotropic response that in turn permits slithering. Further, *C. occipitalis* has been reported to produce $\lambda = 2$ lifting waves [37], which according to our model can generate slithering even in isotropic settings, albeit inefficiently (Fig. 3b). Also interesting is the case of snakes ascending sandy slopes, where sidewinding has been suggested to be an actual necessity to cope with the substrate’s high propensity to deform [27, 35], rather than an advantageous solution as predicted by our model. Both these examples underscore the role of substrate remodeling, reflow, compaction, and associated resistive forces, further enriching limbless locomotion dynamics. A connection between granular media resistive forces [34, 37–40] and our effective–friction framework might further harmonize theory and observations.”*

In this new section we also refer to the reviewer’s suggested literature (Schiebel et al, elife 2020; Shrivastava et al, Science Robotics, 2020; Mazouchova et al, B&B 2013), for completeness.

Finally, we felt compelled to add an additional resource complementing our manuscript: an online, freely-accessible and intuitive sandbox for readers and users to experiment with different actuation and friction conditions, and observe in real time resulting locomotory outputs. We believe this could be an impactful tool for researchers, for the simply curious reader and for outreach activities.

Our sandbox is live at:

https://gazzolalab.github.io/kinematic_snake_sandbox/snake_sandbox.html

The sandbox will be also progressively expanded in time, to encompass more complex and advanced features.

A brief description of this tool can now be found in the Methods section of the manuscript.

Otherwise, I would advocate that this paper be published without delay (provided the authors address the above) -- exciting and cool work!

And PS, I love the optics analogies!

We again thank the reviewer for her/his kind words and enthusiasm for the work, in particular for the optics analogies. We believe the reviewer's comments helped us improve our manuscript as a whole, potentially broadening its appeal across readerships. We hope the reviewer finds the revised manuscript to have addressed the provided comments and suitable for publication.

References used in this response:

- P. E. Schiebel, H. C. Astley, J. M. Rieser, S. Agarwal, C. Hubicki, A. M. Hubbard, K. Diaz, J. R. Mendelson III, K. Kamrin, and D. I. Goldman, *Elife* 9, e51412 (2020).
- N. Mazouchova, P. B. Umbanhowar, & D. I. Goldman, *Bioinspiration & biomimetics* 8, 026007 (2013).
- S. Shrivastava, A. Karsai, Y. O. Aydin, R. Pettinger, W. Bluethmann, R. O. Ambrose, and D. I. Goldman. *Science Robotics* 5, no. 42 (2020).